# Effect of Water Storage and Bleaching on Light Transmission Properties and Translucency of Nanofilled Flowable Composite

**DOI:** 10.3390/ma16010010

**Published:** 2022-12-20

**Authors:** Taghreed Alrefaie, Ahmed Abdou, Waleed Almasabi, Feng Qi, Ayako Nakamoto, Masatoshi Nakajima, Masayuki Otsuki, Yasushi Shimada

**Affiliations:** 1Department of Cariology and Operative Dentistry, Graduate School of Medical and Dental Sciences, Tokyo Medical and Dental University, 1-5-45, Yushima, Bunkyo-ku, Tokyo 113-8549, Japan; 2Prosthodontic Dentistry Department, Division of Biomaterials, Faculty of Dentistry, King Salman International University, El Tur 46512, Egypt

**Keywords:** nanofillers, optical properties, water storage, bleaching

## Abstract

This study investigated the effect of water sorption and bleaching on light transmission properties (Straight-light transmission (G0), Light diffusion (DF) and Amount of transmitted light (AV)) and translucency parameters (TP) of nano-filled flowable composites. A total of 35 composite disks (0.5 mm thickness) were prepared using A2 shade of 5 nanofiller composites (n = 7/each); Beautifil Flow Plus X F03 (SHOFU INC), Clearfil Majesty ES Flow (Kuraray Noritake Dental), Estelite Universal Flow (EUF, Tokuyama Dental), Estelite Flow Quick (Tokuyama Dental) and Filtek Supreme Ultra Flowable Restorative (FSU, 3M ESPE). Then, they were cured by LEDs (VALO, Ultradent) on standard mood (1000 mW/cm^2^) for 20 s. Samples were tested for straight-line transmission (G0), diffusion (DF), the amount of transmitted light (AV) and (TP) immediately after 24 h (dry storage), after 1-week water storage and after each of the three cycles of in-office bleaching (HiLite, SHOFU INC). Result: G0, DF, AV and TP were significantly affected by different materials (*p* < 0.001). The AV of FSU increased significantly after the 1-week water storage, then after the second bleaching cycle (*p* < 0.001). The TP for EUF slightly decreased (*p* = 0.019) after 1-week water storage, then increased throughout bleaching. Conclusion: Ageing/bleaching conditions do not affect G0, DF, AV and TP. The compositional variation between nano-filler composites resulted in a significant difference between materials.

## 1. Introduction

Within the focus of nanotechnology in recent years, composite dental materials have had their share of it, which attracts many researchers and developers to innovate materials for various uses. One such advantage derived from nanotechnology is the usage of nanofillers. Filler types determine the mechanical properties of resin composite materials and reduce the monomer content and polymerization shrinkage, wear, translucency, surface roughness and, thus, polishability, enhancing aesthetics and improving handling properties [1]. Nanofillers are fillers with particle sizes in the 5–100 nm range dispersed into the various polymer matrixes [2]. The filler type, the proportion of inorganic particles, size, and filler distribution significantly influence the resin composite physical, mechanical, and optical properties. As the filler size became small, it demonstrated the highest retention value for composite materials [3]. The more nanofillers were added to the matrix, the more adhesion between them occurred, which avoided the failure of the composite in the early stages. Nanofiller resin composite materials are more resistant to wear with a smoother surface [4]. The aesthetic properties of restorative materials allow them to mimic the natural tooth colour [5]. The light will scatter by filler particles while transmitted within the materials before it merges and reaches the eye. Hence, the colour appearance of the materials is a complex combination of their optical properties [6]. However, even with such a technology, they can still change their optical properties due to eccentric and intrinsic factors.

Fillers affect the optical properties of resin composites through a light incident on them [2]. At the same time, light is either partly reflected from the surface, passing and absorbed or scattered at the surface of the filler and diffused in multiple directions. Consequently, light can transmit through the composite in a straight line or diffused, which got affected by a mismatch filler and a matrix interface [2]. Hence, composites’ light scattering characteristics depend on the filler/matrix interface [2]. Suppose the link between them degrades due to hydrolysis of the silane coupling treatment on the surface of the filler, where the water attracted the free radical hydroxyl group on the filler surface and formed a hydrogen bond [7]. In that case, water will be absorbed, altering the light transmission and translucency of the risen composite [2]. Translucency is essential for dental restorative materials. Most organic matrix and inorganic fillers do not effectively absorb visible light [8]. As a result, the scattering of light might be considered the main reason for low translucency [8]. Using light-emitting diodes (LEDs), in which light intensity is a relevant factor in the polymerization of resin materials, influences the colour stability and microhardness of composite resins [9], which leads to a decrease in water sorption and monomer elution [10].

Bleaching agents improve the esthetics of the natural dentition [11], a non-invasive approach to lighten teeth that are stained extrinsically or intrinsically [8]. Their effect is related to the concentration of whiting gel and exposure time; the longer the exposure, the more significant the colour change and this might cause surface roughness to restorative materials [8]. Composite materials are more prone to surface changes after bleaching [12,13], which affects the risen-filler interface that will lead to differences in light transmission within/from the materials before it reaches the observer’s eyes [6]. This difference in light transmission within/from the materials before and after bleaching will affect the aesthetic appearance of the dental composite.

This study is the first to evaluate the effect of water storage and bleaching on light transmission properties and the translucency of LED-cured nanofiller flowable composites. The null hypothesis investigated is that different ageing and bleaching conditions do not affect the light transmission properties and transparency of different nanofiller resin composites.

## 2. Materials and Methods

### 2.1. Materials

Five commercial tested nanofiller flowable resin composites used in the current study and their composition are listed in Table 1.

### 2.2. Specimen Preparation

Disk-shaped specimens (n = 7/each composite), 6 mm in diameter with 0.5 mm thickness, were prepared on a plastic mould. The flowable resin composites were applied to the mould, covered with a celluloid strip, and polymerized for the 20 s from 0 mm distance with an LED device (VALO, Ultradent); the output light intensity of the curing unit was on standard mood (1000 mW/cm^2^). The LED device intensity was measured using a dental radiometer device (Bluephase Meter II, Ivoclar Vivadent) before the start of the study. The specimen’s upper surface was polished using water-cooled 2000-grit silicon carbide paper and stored in dark-dry conditions for 24 h, followed by 1-week storage in distilled water at 37 °C, then bleaching.

### 2.3. Bleaching Procedure

It uses the HiLite tooth whitening system (35% Hydrogen Peroxide solution, SHOFU INC, Kyoto, Japan). Mix the powder and liquid on the paper dough. Apply the mix while it is turquoise, then leave it on the discs until it changes its colour to white. The cycle lasted for 15 min, was whipped with moist cotton, washed with water, and dried with air. Repeat it for three cycles as instructed by the manufacturer for one dental visit.

Light transmission properties (Straight-light transmission (G0), Light diffusion (DF) and the amount of transmitted light (AV)) and translucency parameters (TP) measurements were performed immediately after polishing, after 24 h, 1-week water storage, first bleaching cycle, second bleaching cycle, and third bleaching cycle.

### 2.4. Measurement of Light Transmission Properties

Using a goniophotometer machine (GP-200, Murakami Color Research Laboratory, Tokyo, Japan) with no filter under standardized conditions (sensitivity: 950; Volume: 522) to measure G0, DF, and AV. When the incidence angle sat at 0°, a two-dimensional distribution graph of transmitted light intensities can obtain from −90° to +90°. While G0 calculate at the peak gain at 0°, DF was calculated using the following Formula (1):(1)$$DF\left[\%\right]=\left[\left[B70{}^{\circ}+B20{}^{\circ}\right]/2]/[B50{}^{\circ}\right]\times 100$$
where *B* is the light intensity at a certain angle. Furthermore, using Image J software (version 1.74 for windows, National Institute of Health, Bethesda, MD, USA), AV was calculated as the total area of the distribution of the light’s graph.

### 2.5. Measurement of Translucency

For CIELAB coordinates L*, a*, and b* measurement, the composite discs were placed on a Black and White background with a reflection spectrophotometer (Crystaleye M639001, Olympus, Tokyo, Japan). Then, through measuring, the colour difference was calculated according to the Formula (2):(2)$$TP={\left[{\left[L_{B}{}^{*}-L_{W}{}^{*}\right]}^{2}+{\left[a_{B}{}^{*}-a_{W}{}^{*}\right]}^{2}+{\left[b_{B}{}^{*}-b_{W}{}^{*}\right]}^{2}\right]}^{1/2}$$

Subscripts *B* and *W* refer to the black and white backgrounds, respectively.

### 2.6. Scanning Electronic Microscopy (SEM)

Ten new discs were prepared for each material described in the Materials and Methods above and stored in dark-dry condition for 24 h. Then, half of the discs (n = 5) were scanned using a scanning electron microscope (JSM-IT100, JEOL, Tokyo, Japan) at a magnification of 6500×. At the same time, the other half (n = 5) was stored in distilled water for 1 week and then we continued with bleaching for three cycles. After that, specimens were scanned with SEM using the same magnification.

### 2.7. Statistical Analysis

Data were checked for normality using the Shapiro–Wilk test and showed non-normal distribution, so the Kruskal–Wallis test was used to compare the tested groups and the ageing/bleaching variable, followed by multiple comparisons with Dunn Bonferroni. A significant level was set at 0.05 (SPSS IBM, Version 26, Armonk, NY, USA).

## 3. Results

### 3.1. Light Transmission Properties

The results of the straight-line transmission (G0) are presented in Table 2, while Table 3 and Table 4 represent diffusion (DF), the amount of transmitted light (AV) results. The Kruskal–Wallis test revealed that G0 & DF were influenced significantly by different materials for all ageing/bleaching conditions (*p* < 0.001), while ageing/bleaching conditions did not affect G0 & DF (*p* > 0.5 for all conditions). For all ageing/bleaching conditions, BFP and ESF exhibited a significantly lower G0, followed by EUF, followed by EFQ and FSU. For the DF, BFP and ESF showed the highest significant values compared to all other materials (*p* < 0.5 for all conditions), while EFQ and FSU showed the lowest DF values for all conditions (*p* < 0.5).

As for AV (Table 4), they were significantly different between all tested resin composites (all, *p* < 0.001) at all ageing/bleaching conditions except 1 week (*p* = 0.063). For ageing/bleaching conditions, there were insignificant changes for all resin composites (*p* > 0.5) except for FSU (*p* < 0.001). The AV values for FSU increased significantly after the 1-week water storage, followed by a further increase after the second bleaching cycle.

### 3.2. Translucency Parameters

The Kruskal–Wallis test revealed that TP (Table 5) was influenced significantly by different materials for all ageing/bleaching conditions (*p* < 0.001). For all ageing/bleaching conditions, EFQ and EUF exhibited the highest significant TP values of the tested composite materials. Different ageing/bleaching conditions had an insignificant effect on all restorative materials (*p* > 0.5) except EUF, which showed a significant decrease in TP values after 1 week followed by an insignificant gradual increase with bleaching cycles.


### 3.3. SEM Observation

Figure 1 shows the surface analysis of tested materials using SEM after 24 h dry storage and after the third bleaching cycle at a magnification of 6.500×. Each material exhibited different filler sizes and shapes.

Beautifil Flow Plus X F03 (BFP) (Figure 1a,b), show a slightly filler detachment from the resin matrix compared to the 24 h dry condition. Clearfil Majesty ES Flow (ESF) (Figure 1c,d) and Estelite Universal Flow (EUF) (Figure 1e,f) show a smooth distribution of fillers even after bleaching. Estelite Flow Quick (EFQ) (Figure 1g,h) shows a spherical regular filler type with slight morphological changes after bleaching. Supreme Ultra Flowable Restorative (FSU) (Figure 1i,j) shows more extensive fillers with different sizes; after bleaching, the fillers detached and revealed cracks in the nanoclusters filler.

## 4. Discussion

With recent technological development, bleaching has become popular, where the patient can have a white smile with a noninvasive approach. However, sometimes patients seeking bleaching might have restored teeth with resin composite [12]. This will lead to direct contact between the bleaching agent and the resin composite; thus, it is essential to understand the effect of bleaching on the resin composite [12]. This study is the first to examine the effect of one visit of office bleaching with 35% hydrogen peroxide after 1-week water storage on degradation of five commercial nano-filled flowable composites using light transmission properties (G0, DF, AV) and TP. The used optical parameters are more sensitive in detecting minor changes that may result after short-term water storage or bleaching [2]. The 1-week water storage significantly increases the AV for FSU and decreases the TP for EUF, while bleaching further increases the AV for FSU after the second cycle and increases the TP for EUF. Therefore, the null hypothesis was partially rejected.

The resin composite tends to absorb water when immersed in it, and the influence of its sorption extent is affected by the composition and volume of the resin matrix, filler load, monomer elution, and curing conditions (type of devices, curing time and distance) [2,10,14,15]. In this study, the 1-week water storage significantly increases the AV of FSU compared to values after 24 h dry storage. Also, EFQ has a higher G0 and AV for ageing conditions than other materials, and their DF values were the lowest. That may be attributed to the lower filler content in EFQ (53 vol %) and FSU (47 vol %). The higher matrix volume resulted in high water sorption [16], which influenced the light transmission properties of FSU and EFQ. Additionally, the degradation link between the filler/matrix interface may reduce the amount of light reflected from the composite surface and its optical properties [2]. On the contrary, EUF and ESF have high filler loads (57 and 59 vol %, respectively). Furthermore, BFP was composed of surface-per-reacted glass-ionomer (S-PRG), which had a high water sorption value after 24 h. However, it decreased after one week [17] due to fluoride release, supporting the diffusion of water whilst not having a substantial value of water absorption [18]. So, EUF, ESF and BFP have a lower extent of water sorption, and their light transmission properties were not significantly affected.

The filler particles are either glass or ceramic; therefore, the effect of hydrogen peroxide could be small or have no effect [13]. In contrast, the organic matrices of resin composite are more prone to surface changes due to repeated application of hydrogen peroxide [12,13]. Bleaching would affect the resin composite due to monomer elution [19,20] and could affect resin matrices and the resin–filler interface [21], which leads to micro-cracks [22,23]. In this study, the light transmission properties and translucency in different material types were due to the composition and volume of the resin matrix and filler types, load, and distribution in resin matrices. EFQ and FSU have lower filler loads among the materials, affecting their light transmission properties after bleaching, increasing the G0 and decreasing the DF non-significantly. However, the AV of FSU increases significantly after the second bleaching cycle due to the effect of hydrogen peroxide on both the organic matrix and the resin–filler interface, which leads to micro-cracks on nanocluster fillers [12,13,21].

BFP, also known as Giomer [24], showed slightly fewer colour changes after bleaching with no static difference than other composite materials [18,25]. Ion-releasing characteristics of S-PRG fillers might buffer the slight acidity of distilled water during storage [16]. Moreover, they might be triggered by the unbalanced pH of the bleaching agent hydrogen peroxide [26] to further dissolute and release more ions that help to buffer the bleaching agent action. While the effect of hydrogen peroxide on ESF and EUF could be small or not affect G0, AV, and DF. The TP for EUF contains silica-zirconia spherical nanoparticles and pre-polymerized fillers composed of an organic polymerizable resin [2], due to the presence of organic polymerizable resin, which will be affected by hydrogen peroxide and increase after the second bleaching cycle.

The roughness in the composite surface was due to matrix degeneration without loss of fillers [27], which might lead to colour changes, which will guide the dentist to replace the restoration due to the colour mismatching with tooth colour [8,28,29]. Nano-filled composite materials possess excellent colour stability and show less surface variation after bleaching [30,31], due to greater filler particles covering a large surface area. The amount of monomer elution will increase or decrease after bleaching [20]. The effect of bleaching agents on the roughness of the material’s surface depends on the type of bleaching agent and the duration of application [22]. Due to the loosely bound between zirconia and silica nanoparticles, which form the nano-clusters filler of FSU, SEM observation of the third bleaching cycle specimen shows micro-cracks within the nano-cluster and their detachment slightly from the resin matrix (Figure 1j). So, it will affect the light scattering, increase the DF, and decrease the TP [2] non-significantly. As for EFQ, SEM shows slight degeneration of the interface between the resin matrix and silica-zirconia spherical fillers (Figure 1h), which revealed no significant changes in light transmission properties and TP. This difference might be due to EFQ’s spherical and regular filler shape compared to FSU. SEM images for BFP (Figure 1b) show the changes due to (S-PRG) that got affected by the bleaching agent [23]. However, there were no changes in light transmission properties and TP due to their excellent natural shade reproduction due to ion release [18]. ESF contains nano-cluster fillers composed of silica nanoparticles treated with a highly hydrophobic silane coupling agent, and we assume that it is the cause of their high resistance to hydrolytic and chemical stress during water storage [2] and bleaching. Moreover, EUF contains pre-polymerized fillers resistant to external stresses, which leads them to have no morphological alteration [2] (Figure 1d,f, respectively).

To sum up, the light transmission characteristic which affects the TP is influenced by filler load, type, size and shape [32]. The higher the filler load, the lesser the polymerization shrinkage of the resin composite [33]. Each filler’s particles have different optical properties, leading to different light scattering on the composite risen, which affects the curing depth and polymerization shrinking in addition to the light curing time [34]. The light cure unit affects the amount of monomer elution, affecting the water sorption [9]. The office bleaching for one dental visit does not induce a detectable change in the optical properties of nanofiller composite or giomer despite the surface changes [25]. The results of different studies might relate to differences in methodology, type and concentration of bleaching agents, composite types [23] and the light cure method.

The polishing and finishing processes of the specimen might have an impact on the surface roughness of the evaluated parameters, which could be a restriction. However, we have standardized the polishing step for all specimens by using #2000-grit silicon carbide paper. Further research is necessary with different concentrations of bleaching agents and ageing conditions to characterize optical and aesthetic properties of resin composite materials.

## 5. Conclusions

In conclusion, water storage and bleaching will increase the amount of light transmission and decrease the amount of light scattering for low filler load nano-filled flowable composite. On the other hand, light transmission properties and translucency for high filler load nano-filled composite were more stable under similar conditions.

## Figures and Tables

**Figure 1 materials-16-00010-f001:**
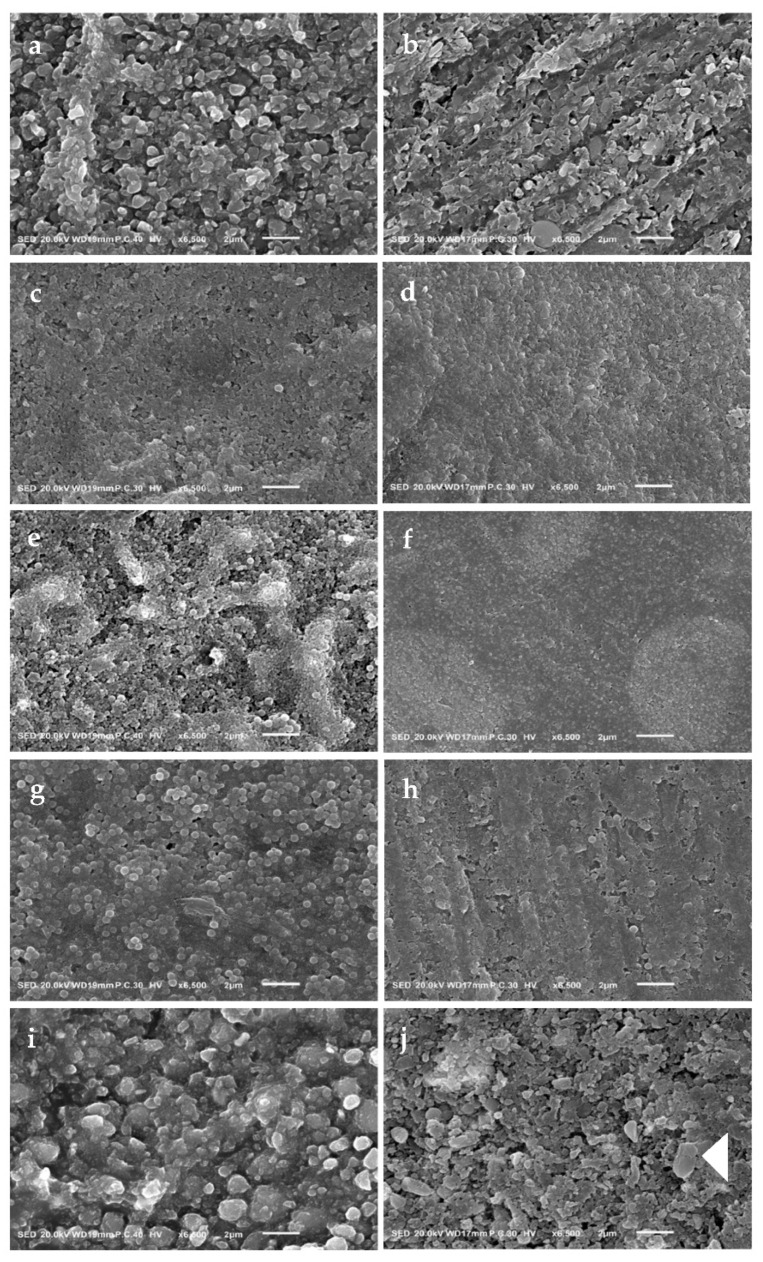
SEM images of resin composite surface (magnification 6500×). The left column shows the surface after 24 h of dry conditions, while the right column shows the surface after the third bleaching cycle. (**a**,**b**) BFP. (**c,d**) ESF. (**e**,**f**) EUF. (**g**,**h**) EFQ. (**i**,**j**) FSU, while the arrow in (**j**) shows the crack in nanocluster filler particles.

**Table 1 materials-16-00010-t001:** Composition of tested materials according to the manufacturer’s data.

Materials (Shade, Abbreviation)	Manufacturer (Batch Number)	Filler Composition	Resin Matrix	Filler Content (wt %–vol %)
**Beautifil Flow Plus X F03 (A2, BFP)**	SHOFU INC, Kyoto, Japan (042019)	S-PRG ^1^ filler based on fluoroboroaluminosilicate glass (400 nm), Polymerization initiator, Pigments and others	Bis-GMA ^2^, TEGDMA ^3^, Bis-MPEPP ^4^	66.8 wt %—NA
**Clearfil Majesty ES Flow (A2, ESF)**	Kuraray Noritake Dental, Tokyo, Japan (CD0307)	Silanated barium glass filler, Silanated silica nanocluster filler (0.18–3.5 µm)	TEGDMA ^3^, hydrophobic aromatic dimethacrylate	75 wt %–59 vol %
**Estelite Universal Flow (A2, EUF)**	Tokuyama Dental, Tokyo, Japan (0421)	Spherical silica-zirconia filler (200 nm), Prepolymerized filler (200 nm)	Bis-GMA ^2^, Bis-MPEPP ^4^, TEGDMA ^3^, UDMA ^5^	71 wt %–57 vol %
**Estelite Flow Quick (A2, EFQ)**	Tokuyama Dental (J069)	Silica-zirconia supra-nano spherical filler (0.07 µm, 0.4 µm)	Bis-MPEPP, TEGDMA ^3^, UDMA ^5^	71 wt %–53 vol %
**Filtek Supreme Ultra Flowable Restorative (A2, FSU)**	3M, St. Paul, MN, USA (NC10483)	Silica (20 nm, 75 nm), Zirconia (5–10 nm), Zirconia/silica clusters (0.6–10µm), Ytterbium fluoride (0.1–5 µm)	Bis-GMA ^2^, TEGDMA ^3^, Procrylat resin^6^	65 wt %–47 vol %

^1^ surface pre-reacted glass-ionomer, ^2^ bisphenol A-glycidyl methacrylate, ^3^ triethyleneglycol dimethacrylate, ^4^ bisphenol A polyethoxy methacrylate, ^5^ urethane dimethacrylate, ^6^ [2,2-bis[4-[3methacryloxypropoxy]phenyl]propane], NA: not available.

**Table 2 materials-16-00010-t002:** Straight-light transmission (G0) result.

G0	Imm	24 h	1 w	1B	2B	3B
**BFP**	5.82 ^aB^ ± 0.71	5.93 ^aB^ ± 0.66	5.79 ^aB^ ± 0.73	5.89 ^aB^ ± 0.6	5.7 ^aB^ ± 0.66	5.6 ^aB^ ± 0.6
**ESF**	6.43 ^aB^ ± 0.4	6.21 ^aB^ ± 0.49	6.29 ^aB^ ± 0.42	6.24 ^aB^ ± 0.4	6.15 ^aB^ ± 0.39	6.04 ^aB^ ± 0.5
**EUF**	9.16 ^aC^ ± 0.8	9 ^aC^ ± 0.79	9.11 ^aC^ ± 1.41	8.58 ^aC^ ± 0.77	8.36 ^aC^ ± 0.55	8.16 ^aC^ ± 0.53
**EFQ**	14.99 ^aA^ ± 1.58	17.2 ^aA^ ± 3.94	15.63 ^aA^ ± 1.96	15.17 ^aA^ ± 2	17.21 ^aA^ ± 4.38	17.3 ^aA^ ± 2.55
**FSU**	15.62 ^aA^ ± 4.49	15.96 ^aA^ ± 3.6	18.38 ^aA^ ± 2.42	17.37 ^aA^ ± 3.79	16.81 ^aA^ ± 4.06	16.19 ^aA^ ± 3.51

Different Uppercase superscript letters indicate significant differences within the composites column; lowercase superscript letters indicate significant differences within a row (Time/Bleaching). Imm: immediately, 24 h: 24 h dry storage, 1 w: 1 week water storage, 1B: 1st bleaching cycle, 2B: 2nd bleaching cycle, 3B: 3rd bleaching cycle, BFP: Beautifil Flow Plus X F03, ESF: Clearfil Majesty ES Flow, EUF: Estelite Universal Flow, EFQ: Estelite Flow Quick, FSU: Filtek Supreme Ultra Flowable Restorative.

**Table 3 materials-16-00010-t003:** Light diffusion (DF) result.

DF	Imm	24 h	1 w	1B	2B	3B
**BFP**	75.09 ^aA^ ± 5.17	74.41 ^aB^ ± 5.34	76.94 ^aA^ ± 4.71	76.42 ^aA^ ± 4.98	77.56 ^aA^ ± 3.24	75.94 ^aA^ ± 3.78
**ESF**	71.39 ^aA^ ± 2.68	73.85 ^aB^ ± 3.02	71.23 ^aA^ ± 3	71.63 ^aA^ ± 2.56	72.53 ^aA^ ± 2.43	68.67 ^aA^ ± 3.35
**EUF**	59.06 ^aB^ ± 3.26	60.02 ^aC^ ± 3.54	57.58 ^aB^ ± 5.53	59.52 ^aB^ ± 3.51	60.15 ^aB^ ± 2.99	60.47 ^aB^ ± 3
**EFQ**	43.2 ^aC^ ± 4.92	43.06 ^aA^ ± 5.68	39.84 ^aC^ ± 5.8	39.27 ^aC^ ± 3.69	39.99 ^aC^ ± 3.79	39.27 ^aC^ ± 6.03
**FSU**	42.28 ^aC^ ± 5.18	35.97 ^aD^ ± 2.99	37.04 ^aC^ ± 3.6	37.15 ^aC^ ± 2.43	39.04 ^aC^ ± 3.57	39.55 ^aC^ ± 4.56

For the interpretation of significant differences and the explanation of abbreviations, please refer to Table 2.

**Table 4 materials-16-00010-t004:** The amount of transmitted light (AV) result.

AV	Imm	24 h	1 w	1B	2B	3B
**BFP**	18546.429 ^aB^ ± 1971.098	19382.286 ^aB^ ± 1130.772	19179.571 ^aA^ ± 1818.38	19527 ^aB^ ± 1241.322	19399.571 ^aB^ ± 1542.524	18638.571 ^aB^ ± 1069.993
**ESF**	20155 ^aB^ ± 1673.716	20468.857 ^aB^ ± 917.085	20282 ^aA^ ± 940.799	20283.429 ^aB^ ± 1013.792	20041.429 ^aB^ ± 987.606	19086.143 ^aB^ ± 1118.253
**EUF**	24934.286 ^aA^ ± 1757.922	25853.571 ^aA^ ± 1366.025	24159 ^aA^ ± 3462.964	24096.286 ^aA^ ± 1332.973	24020.714 ^aA^ ± 977.931	23637 ^aA^ ± 828.356
**EFQ**	20390.571 ^aB^ ± 1800.671	19077.429 ^aB^ ± 1374.083	20935.143 ^aA^ ± 1426.527	19405.571 ^aB^ ± 1814.682	20462.714 ^aAB^ ± 2570.65	21650.143 ^aAB^ ± 2259.822
**FSU**	17821.143 ^aB^ ± 826.717	18101.143 ^abB^ ± 3165.314	20188.143 ^abA^ ± 1737.958	20982 ^abAB^ ± 2549.037	23759.714 ^bAB^ ± 2614.292	24010.857 ^bA^ ± 2561.184

For the interpretation of significant differences and the explanation of abbreviations, please refer to Table 2.

**Table 5 materials-16-00010-t005:** Translucency parameters (TP) result.

TP	Imm	24 h	1 w	1B	2B	3B
**BFP**	24.413 ^aB^ ± 0.706	23.535 ^aB^ ± 0.462	23.93 ^aC^ ± 0.676	24.373 ^aA^ ± 0.521	24.204 ^aA^ ± 0.414	24.143 ^aA^ ± 0.3
**ESF**	25.87 ^aB^ ± 0.797	25.441 ^aA^ ± 0.768	25.07 ^aC^ ± 1.175	25.442 ^aA^ ± 0.975	25.694 ^aB^ ± 0.758	25.193 ^aA^ ± 1.023
**EUF**	28.86 ^aAC^ ± 1.035	28.768 ^aC^ ± 1.051	27.389 ^bB^ ± 1.06	27.24 ^bB^ ± 1.006	27.953 ^abC^ ± 1.117	28.151 ^abB^ ± 0.988
**EFQ**	28.134 ^aC^ ± 0.753	28.593 ^aC^ ± 1.076	27.598 ^aAB^ ± 0.87	27.676 ^aB^ ± 0.959	28.496 ^aC^ ± 1.002	28.726 ^aB^ ± 0.792
**FSU**	26.158 ^aBC^ ± 1.331	25.302 ^aAB^ ± 1.245	25.222 ^aC^ ± 1.127	25.157 ^aA^ ± 0.854	25.008 ^aAB^ ± 0.758	24.276 ^aA^ ± 0.579

For the interpretation of significant differences and the explanation of abbreviations, please refer to Table 2.

## Data Availability

Original data can be requested from corresponding authors (T.A and A.A.) upon reasonable request.

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
