# Peer review of "Effect of Water Storage and Bleaching on Light Transmission Properties and Translucency of Nanofilled Flowable Composite"

_materials, 2022, doi:10.3390/ma16010010_

Round 1

Reviewer 1 Report

I read an interesting article with great interest, but as a reviewer I have some comments:

Abstract

 When to use the abbreviations for the first time ((G0, DF and AV) it is necessary to explain what they mean.

in-office bleaching- if, you provide the trade names of composite materials in the abstract, then also provide the name and the manufacturer of the whitening agent

 introduction

Line 65

Hence, composites' light scatteing characteristics depend on the filler/matrix interface (24). If the link between them degrades, water will be absorbed, altering the light transmission and translucency of risen composite -it would be good to add 1-2 sentences why the degradation of the composite material occurs due to the hydrolysis of the silanes that are on the surface. Such problem has been clarified e.g. in the work of

 Raszewski Z, Nowakowska-Toporowska A, Nowakowska D, WiÄ™ckiewicz W. Update on Acrylic Resins Used in Dentistry. Mini Rev Med Chem. 2021;21(15):2130-2137. doi: 10.2174/1389557521666210226151214. PMID: 33634758.

Zanchi, C.H., Ogliari, F.A., Marques e Silva, R. et al. Effect of the silane concentration on the selected properties of an experimental microfilled composite resin. Appl Adhes Sci 3, 27 (2015). https://doi.org/10.1186/s40563-015-0054-0

Materials and methods

Line 87

output light intensity of the curing unit was on standard mood (1000mW/cm2)- how do you know that the mile lamp is so powerful? Did you measure it with a radiometer?

Line 108

DF(%)  =((B70o+B20o)/2)/B5o )X100, where B is the light intensity at a certain angle

-          for equations please use equations editor otherwise it's unreadable. Maybe you can write: Diffusion was calculate using the equation 1.

-            Give below the equation with 1.

-          The scientific article does not use the personal forum we, I, but writeit impersonally

Line 117

 Please use equation editor

Line 122

-              The scientific article does not use the personal forum we, I, but writeit impersonally: it was measured, it was tested,  it was calculated ......

Results

you use abbreviations a lot, maybe instead of repeating them every second sentence you might use an abbreviation in one and the full name in the other. straight-line transmission (G0), diffusion (DF), the amount of transmitted light (AV).

SEM observation

Maybe it is good to give at the very beginning the names of the composite used and not abbreviations in the discussion of the results, it is a bit difficult to understand when reading it for the first time.

Table 1

How do you know the size of the particles and quantity of the filler in these composite materials, whether it is written in the manual, advertising poster or was it measured by you. This requires an explanation?

Table 2, Table 3 Table 4

light transmission seems to have some unit - it's good to enter [%]. In the title of Table 2, Table 3 and Table 4

Discussion

While BFP, also known as Giomer (14), has excellent natural shade reproduction due to ion release-

It seems to me that ionizing the ions will not have a good effect on the color of the material, there is hydrolysis, the filler becomes more white, the color changes. I have already tested this in various materials.

Referencwes

32. , 29    26, 27Feridun Hürmüzlü, Vahti Kılıç. Analysis of Monomer Elution from Bulk-fill and Nanocomposites Cured  with Different Light Curing Units Using High Performance Liquid Chromatography. Journal of Photopolymer Science and Technology Volume 33, Number 1 (2020) 27-36.- if we give the full surname and the first letter of the first name, it is everywhere in every reference

24. 19....surname capital letter first only, then first letter of the first name. Literature must be equally presented everywhere

18- if the year is given everywhere without parentheses, then position 18 is the same?

Thank you for the opportunity to be a reviewer of this article and good luck with your further research

Author Response

Reviewer 1

Point 1 “When to use the abbreviations for the first time ((G0, DF and AV) it is necessary to explain what they mean.

in-office bleaching- if, you provide the trade names of composite materials in the abstract, then also provide the name and the manufacturer of the whitening agent

Thank You for your valuable comment. We have added the explanation for the abbreviations and the manufacturer name were added.

Point 2 “Line 65

Hence, composites' light scatteing characteristics depend on the filler/matrix interface (24). If the link between them degrades, water will be absorbed, altering the light transmission and translucency of risen composite -it would be good to add 1-2 sentences why the degradation of the composite material occurs due to the hydrolysis of the silanes that are on the surface

Thank You for your valuable comment. We have added the explanation as advised.

Point 3 “Line 87

output light intensity of the curing unit was on standard mood (1000mW/cm2)- how do you know that the mile lamp is so powerful? Did you measure it with a radiometer?

Thank You for your meticulous revision. Yes we have confirmed the LED intensity before the work and we have added that in the text.

Line 108

DF(%)  =((B70o+B20o)/2)/B5o )X100, where B is the light intensity at a certain angle

-          for equations please use equations editor otherwise it's unreadable. Maybe you can write: Diffusion was calculate using the equation 1.

-            Give below the equation with 1.

Thank You for your comment. We have used equation editor as advised in the text.

-          The scientific article does not use the personal forum we, I, but writeit impersonally

Line 117

We apologize for the mistakes. We have corrected that in the manuscript.

 Please use equation editor.

Line 122

Thank You for your comment. We have used equation editor as advised in the text.

-              The scientific article does not use the personal forum we, I, but writeit impersonally: it was measured, it was tested,  it was calculated ......

We apologize for the mistakes. We have corrected that in the manuscript.

Point 3 “you use abbreviations a lot, maybe instead of repeating them every second sentence you might use an abbreviation in one and the full name in the other. straight-line transmission (G0), diffusion (DF), the amount of transmitted light (AV).

We apologize for the confusion. We have tried to follow your recommendation in the manuscript.

SEM observation

Maybe it is good to give at the very beginning the names of the composite used and not abbreviations in the discussion of the results, it is a bit difficult to understand when reading it for the first time.

Thank You for your comment. We have corrected the text.

Point 4 “Table 1

How do you know the size of the particles and quantity of the filler in these composite materials, whether it is written in the manual, advertising poster or was it measured by you. This requires an explanation?

Thank You for your comment. The data used was manufacturer data, and we have corrected the text.

Table 2, Table 3 Table 4

light transmission seems to have some unit - it's good to enter [%]. In the title of Table 2, Table 3 and Table 4’’

Thank You for your comment. Table 1 shows that all the material's proportions were written according to manual instructions.

As for units for G0, AV and DF, according to the reference, only DF have the unit %, while for both G0 and AV, we found no reference for their unit.

Point 5 ‘’ While BFP, also known as Giomer (14), has excellent natural shade reproduction due to ion release-

It seems to me that ionizing the ions will not have a good effect on the color of the material, there is hydrolysis, the filler becomes more white, the color changes. I have already tested this in various materials.’’

We apologize for the confusion. The sentence was adjusted with reference for more clearance.

Point 6 ‘’ 32. , 29    26, 27Feridun Hürmüzlü, Vahti Kılıç. Analysis of Monomer Elution from Bulk-fill and Nanocomposites Cured  with Different Light Curing Units Using High Performance Liquid Chromatography. Journal of Photopolymer Science and Technology Volume 33, Number 1 (2020) 27-36.- if we give the full surname and the first letter of the first name, it is everywhere in every reference

  1. 19....surname capital letter first only, then first letter of the first name. Literature must be equally presented everywhere

18- if the year is given everywhere without parentheses, then position 18 is the same?

We apologize for the mistakes. We have corrected the references in the manuscript.

Reviewer 2 Report

Dental composite resins are the most commonly used dental materials, and their advantages include relatively favorable aesthetic properties. The work concerns the influence of bleaching on this type of properties, so it is primarily practical.

·         I have some suggestions of improvements.

·         The introduction does not sufficiently explain why it was decided to address this issue. Aesthetic properties have been described to briefly, the previous work on the effect of bleaching on aesthetics has been insufficiently discussed.

·         What shade/shades of materials was used? Is it important for results?

·         “The specimen's upper surface was polished using water-cooled 2000-grit silicon carbide paper” - what roughness was obtained after polishing? Was it similar to clinical conditions? You should discuss this limitation because it may influence esthetic properties.

·         Give the references in the methodology.

·         “while the arrow in b shows the crack in nanocluster filler particles. “, should be in “d”. Moreover, it is not very visible in this photo...

·         During experiment you measured CIELAB coordinates, I think that this work will be more complete it you show additionally ΔE and if  it will be justified also  ΔL, Δb, Δb changes. Work will be more complete.

·         In my opinion your divagations in lines 208-223 are not fully justified. You didn’t measure water sorption and you don’t have any prove that there were differences in water sorption. It is only you supposition . There is also no evidence that storage in water (in contrast to dry conditions) caused the changes as you didn't test after 1-week storage in dry conditions. So it might as well have been changes simply occurring in the material. Please rearrange this paragraph to distinguishes speculation and consideration from fact.

·         In table 1, you give the mass concentrations of the filler, and in the discussion, the volume concentrations. How did you do the conversions? Where did these numbers come from?

·         Measurements of aesthetic features are primarily of practical importance. Therefore, please write in the discussion what is the practical meaning of the particular measured properties. What values are expected, desired. When are changes clinically important?

Author Response

Reviewer 2

Point 1 “The introduction does not sufficiently explain why it was decided to address this issue. Aesthetic properties have been described to briefly, the previous work on the effect of bleaching on aesthetics has been insufficiently discussed.”

Thank you for your comment. We agree with you. An explanation was added with the explanation of light transmission properties and their reflection and translucency parameters.

Point 2” What shade/shades of materials was used? Is it important for results?”

Thank you for your comment. We unified the shade for all composites to be A2, as mentioned in Table 1.

Point 3 “The specimen's upper surface was polished using water-cooled 2000-grit silicon carbide paper” - what roughness was obtained after polishing? Was it similar to clinical conditions? You should discuss this limitation because it may influence esthetic properties.”

Thank you for your comment, and we apologize for the confusion. We agree with the reviewer’s comment. Polishing with 2000-grit silicon carbide was used first to standardize the surface of all the specimen. Seconed, because the used method of evaluation might be affected with the roughness of the specimen so 2000-grit was selected.

Point 4 “    Give the references in the methodology”

Thank you for your comment. Reference was added to the methodology.

Point 5 “while the arrow in b shows the crack in nanocluster filler particles. “, should be in “d”. Moreover, it is not very visible in this photo...”

The arrow in pic j I will add the pic in a different attachment.

Point 6 “During experiment you measured CIELAB coordinates, I think that this work will be more complete it you show additionally ΔE and if  it will be justified also  ΔL, Δb, Δb changes. Work will be more complete.”

Thank you for your comment. We agree with you; however, the work is related to the degradation of the composite rather than the variation in colour after the bleaching. So our tested parameters will give a good explanation of the composite degradation.

Point 7 ‘’  In my opinion your divagations in lines 208-223 are not fully justified. You didn’t measure water sorption and you don’t have any prove that there were differences in water sorption. It is only you supposition . There is also no evidence that storage in water (in contrast to dry conditions) caused the changes as you didn't test after 1-week storage in dry conditions. So it might as well have been changes simply occurring in the material. Please rearrange this paragraph to distinguishes speculation and consideration from fact.”

Thank you for your comment. Water storage and water sorption of resin composites are well documented in the literature. We tried to explain our variation of results using water sorption as one factor that may affect our results. We have added a reference to support our explanation.

Rummani G, Ide K, Hosaka K, Tichy A, Abdou A, Otsuki M, Nakajima M. Regional ultimate tensile strength and water sorption/solubility of bulk-fill and conventional resin composites: The effect of long-term water storage. Dental Materials Journal. 2021 Nov 25;40(6):1394-402.

Giannini M, Di Francescantonio M, Pacheco RR, Boaro LC, Braga RR. Characterization of water sorption, solubility, and roughness of silorane-and methacrylate-based composite resins. Operative dentistry. 2014;39(3):264-72.

Point 8 “  In table 1, you give the mass concentrations of the filler, and in the discussion, the volume concentrations. How did you do the conversions? Where did these numbers come from?”

Thank you for your comment, and we apologize for the confusion. Some manufacturers reported both the Vol% and wt%. One manufacturer didn’t mention the volume percentage. So we used the wt% in the composition table for easy comparison between the materials. But for discussion part we used the Vol% which is more informative than wt% for the explanations.

Point 9 “Measurements of aesthetic features are primarily of practical importance. Therefore, please write in the discussion what is the practical meaning of the particular measured properties. What values are expected, desired. When are changes clinically important?”

Thank you for your comment, and we apologize for the confusing. The aim of our research is related to the degradation of composite, which was evaluated using changes in optical parameters and not related to the aesthetic characteristic of the tested resin composites. We clarified the text for easier understanding in the discussion.

Reviewer 3 Report

In the study, the effect of bleaching and water storage on light transmission properties and the translucency of nanofiller flowable composites has been investigated. However they should explain the question before acceptance.

1.     Authors should explain the validation method for their measurement.

2.     Authors must report the effect of bleaching and water storage on color of composite.

3.     There is no data to show the flowability of the nano filler.

4.     Is there any data about water absorption by the composite?

5.     Author should explain and evaluate the opalescence characteristic after exposure.

6.     In discussion section, authors should compare their results with recent articles in the field.

7.     Author must address the novelity of the work in comparison to published article.

Author Response

Reviewer 3
Point 1 “Authors should explain the validation method for their measurement.”
Thank you for your comment. We have explained the methods used in discussion with suitable references.

Point 2” Authors must report the effect of bleaching and water storage on color of composite.”
Thank you for your comment, and we apologize for the confusion. The aim of our research is related to the degradation of composite, which was evaluated using changes in optical parameters and not related to aesthetic or colour characteristics of the tested resin composites. We clarified the text for easier understanding in the discussion.

Point 3 “There is no data to show the flowability of the nano filler.”
Thank you for your comment, and we apologize for the confusion. We didn’t test the flowability of the resin composite tested. We have selected commercial products which follow similar standards. 

Point 4 “  Is there any data about water absorption by the composite?”
Thank you for your comment. Water storage and water sorption of resin composites are well documented in the literature. We tried to explain our variation of results using water sorption as one factor that may affect our results. We have added a reference to support our explanation.

Rummani G, Ide K, Hosaka K, Tichy A, Abdou A, Otsuki M, Nakajima M. Regional ultimate tensile strength and water sorption/solubility of bulk-fill and conventional resin composites: The effect of long-term water storage. Dental Materials Journal. 2021 Nov 25;40(6):1394-402. 

Giannini M, Di Francescantonio M, Pacheco RR, Boaro LC, Braga RR. Characterization of water sorption, solubility, and roughness of silorane-and methacrylate-based composite resins. Operative dentistry. 2014;39(3):264-72.

Point 5 “Author should explain and evaluate the opalescence characteristic after exposure.”
Thank you for your comment, and we apologize for the confusion. Please refer to the point 2 response.

Point 6 “   In discussion section, authors should compare their results with recent articles in the field.”
Thank you for your comment. The current study is the first to explain the effect of bleaching on flowable composites using optical parameters to detect a minor degradation effect. We have added that to the discussion. So the discussion is limited to explaining the results rather than a comparison with other articles, as the data is limited for comparisons. 

Point 7 “Author must address the novelity of the work in comparison to published article.”
Thank you for your comment. The current study is the first to explain the effect of bleaching on flowable composites using optical parameters to detect the degradation's minor effect. We have added that to the discussion.

Round 2

Reviewer 2 Report

The authors did not make the suggested changes to the manuscript sufficiently. Explanations are vague and corrections in the work are very minor. Please do it more carefully.

Author Response

Thank you very much for your comments.

Kindly see the attachment 

Reviewer 3 Report

Dear Editors

The authors have been addressed the most requirements for publication.

Author Response

Thank you very much for your comments.